# Disdyakis Triacontahedron DGGS

**John Hall \*, Lakin Wecker, Benjamin Ulmer and Faramarz Samavati**

Department of Computer Science, University of Calgary, Calgary AB T2N 1N4, Canada;
lakin.wecker@ucalgary.ca (L.W.); blulmer@ucalgary.ca (B.U.); samavati@ucalgary.ca (F.S.)
\* Correspondence: jshall@ucalgary.ca; Tel.: +1-403-473-2659

**Abstract:** The amount of information collected about the Earth has become extremely large. With this information comes the demand for integration, processing, visualization and distribution of this data so that it can be leveraged to solve real-world problems. To address this issue, a carefully designed information structure is needed that stores all of the information about the Earth in a convenient format such that it can be easily used to solve a wide variety of problems. The idea which we explore is to create a Discrete Global Grid System (DGGS) using a Disdyakis Triacontahedron (DT) as the initial polyhedron. We have adapted a simple, closed-form, equal-area projection to reduce distortion and speed up queries. We have derived an efficient, closed-form inverse for this projection that can be used in important DGGS queries. The resulting construction is indexed using an atlas of connectivity maps. Using some simple modular arithmetic, we can then address point to cell, neighbourhood and hierarchical queries on the grid, allowing for these queries to be performed in constant time. We have evaluated the angular distortion created by our DGGS by comparing it to a traditional icosahedron DGGS using a similar projection. We demonstrate that our grid reduces angular distortion while allowing for real-time rendering of data across the globe.

**Keywords:** discrete global grid systems; digital Earth; equal-area projection; geometric modeling

---

## 1. Introduction

In the modern era, we are increasingly collecting vast amounts of information about the Earth; many zettabytes of data are already available, and more is being collected daily [1,2]. The use of this data ranges from sophisticated experiments and analysis to daily use by the public to ensure their decisions are well informed. In order to perform these tasks, data usually needs to be combined across varying attributes and scales. For example, combining high-resolution global weather patterns, the position and geometry of business assets and the likelihood of political instability to assess insurance risk. Thus, we need a tool for integrating, analyzing, simulating, distributing and visualizing this data so that it can be leveraged to solve real-world problems [2].

As a result, the idea of a Digital Earth (DE) was pursued. In the early stages of DE, many ideas from traditional cartography were extended to a digital setting, and virtual representations of traditional flat maps became the standard [3]. In a map-based DE, the Earth data is projected to a flat map using any one of the multiple taxa of map projections, many of which are still in use today [4]. Analysis can then be performed in the planar setting using conventional Euclidean methods. However, projecting the Earth to a flat map introduces significant and unavoidable distortion, which introduces errors in analysis [5].

Globes are an alternative to flat maps that produce far less distortion [3]. While physical globes are not easily scaled and difficult to manufacture, a DE provides an opportunity to address some of the shortcomings of physical globes [3]. With a globe, analysis can be performed directly in spherical space producing accurate results. However, these analyses may be prohibitively computationally expensive.

In some cases, techniques may not even exist to do the analysis directly in spherical space. Because of this, many DE systems still use single flat maps as their underlying representation [3].

Motivated by the desire to reduce distortion and faced with the challenge of efficient computation, polyhedral approximations of the Earth have been explored. Approximating the surface of the Earth with a convex polyhedron does not completely eliminate distortion, but significantly reduces it while allowing each face of the polyhedron to be treated as a flat map [6]. More faces in the polyhedron allow for a closer approximation of the Earth and a further reduction in distortion [6].

A Discrete Global Grid System (DGGS) is a polyhedron based DE where the surface of the Earth is discretized into mostly regular, multi-resolution cells. These cells are uniquely indexed and used as placeholders for geospatial data associated with the corresponding location on the Earth. A DGGS is generally comprised of four parts:

1. Initial Polyhedron: The coarsest discretization of the Earth into cells
2. Projection: A mapping between each point on the Earth and a corresponding point on the polyhedron (see Figure 1)
3. Indexing: Assigning a unique identifier to each cell
4. Refinement: A way to hierarchically subdivide coarse cells into finer ones creating multi-resolution cells

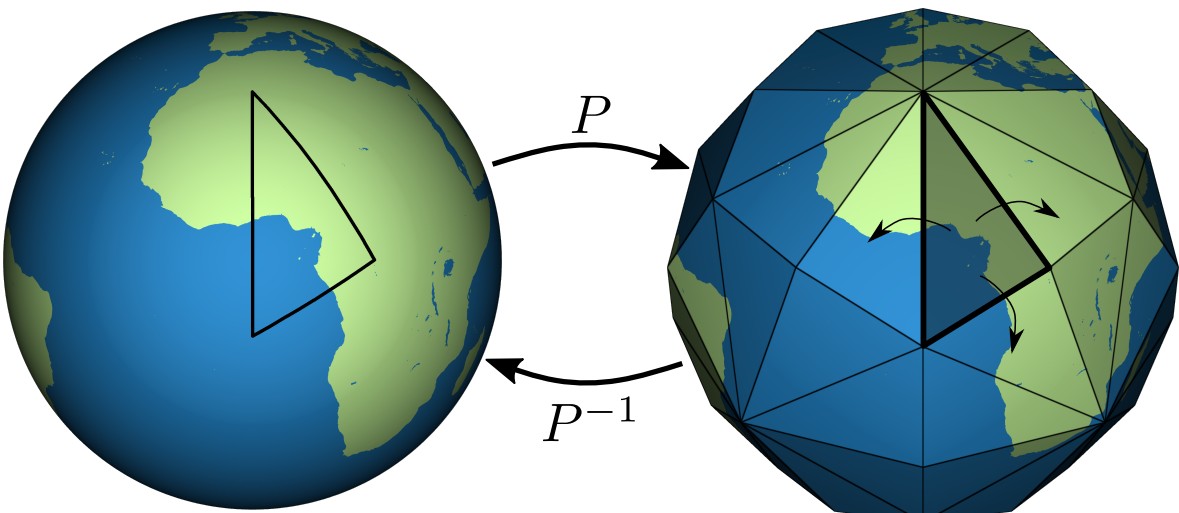

**Figure 1.** Projection between the polyhedron and the Earth. Mirror symmetry is present along the edges of all adjacent triangles in the Disdyakis Triacontahedron (DT).

Despite a considerable amount of research into how to best choose these components, every DGGS is developed with certain properties in mind with tradeoffs made to achieve specific goals. One such property is that the DGGS cells should have equal area at all supported resolutions. Additionally, the cells should be as compact as possible (roughly should take up a large area of their bounding circle). In combination, these two properties allow the cells to represent geospatial data more accurately. The DGGS should also allow for efficient operations, including point queries, neighbourhood queries and hierarchy traversal. In conjunction with the choice of the initial polyhedron, a projection (and its inverse) is needed from the sphere to this polyhedron. The projection chosen should preserve the above properties for cells of the polyhedron and their spherical counterparts at all resolutions. It should also reduce the distortion created in the resulting planar cells as much as possible.

The choice of which initial polyhedron to use is motivated by many different factors and has led to multiple lists of ideal properties to satisfy [7,8]. Platonic solids and some Archimedian solids, such as the truncated icosahedron, are currently used as the initial polyhedron for constructing DGGS [2]. The Platonic solids are good choices because the faces are equal area and compact.

Additionally, the symmetry and regularity of these polyhedra allow for efficient queries. For most of the platonic solids, several projections have been developed that preserve the equal-area criterion. These projections are typically constructed by a piecewise mapping carefully defined on each face of the initial polyhedron. While being able to preserve area, it is impossible to simultaneously remove angular and shape distortion (see Figure 2). The amount of distortion produced depends not only on the type of piecewise mapping but also on how closely the faces of the polyhedron approximate the surface of the Earth. Therefore, it is advantageous to choose a polyhedron with more faces that will reduce distortion [6]. Another important characteristic of DGGS projections is their computational cost, as many important DGGS operations utilize the projection. For example, in order to render data across the surface of the Earth, the data must be inverse projected from the polyhedral data structure where it is stored to its position on the globe. All points of the geometry being used to render the globe need this operation performed to produce an accurate rendering, which necessitates that this operation is efficient. These projections can be a significant bottleneck within a DGGS, and work has been done to increase their speed [9,10]. An important challenge for DGGS research is to find new initial polyhedra and associated area-preserving projections that further reduce angle and shape distortion while maintaining fast and efficient DGGS operations.

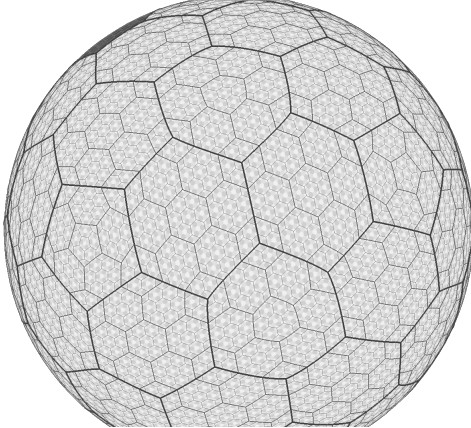

**Figure 2.** Snyder Equal Area projection performed on a regular hexagon grid [11]. While the areas of the projected hexagons are preserved, the shapes and angles are not. Image modified from [12], licensed under Creative Commons BY-SA 3.0.

To address this challenge, we introduce a new DGGS that uses a Disdyakis Triacontahedron (DT) as the initial polyhedron (see Figure 1). The DT is a Catalan solid with 120 identical triangular faces where all vertices lie on a sphere. This polyhedron was chosen because it has the most faces of any Catalan solid. In this convex, spherical polyhedron, all adjacent faces are identical mirror images of each other along their adjoining edge, and the faces are compact. When the initial polyhedron is changed to a DT, other components of the DGGS must be revisited and potentially modified. The DT face shapes are different from the shapes in platonic solids, and existing projections either do not work or must be modified to accommodate this change. For example, Snyder equal-area projection can be used for the truncated icosahedron, but it expects regular pentagonal and hexagonal faces which are not present in the DT [13]. To create an equal-area projection for the DT, we modify the equal-area vertex-oriented great circle projection proposed by van Leeuwen and Strebe [14]. We use 1:4 longest edge bisection triangle refinement to conserve the compactness of the triangles during refinement [15]. To support efficient operations, we develop an indexing scheme inspired by Atlas of Connectivity Maps (ACM) [16]. Using some simple modular arithmetic, we address geospatial queries on the grid with constant time algorithms and implement important operations such as neighbourhood queries, parent/child relationships and point queries.

In order to evaluate our DGGS, we compare the angular distortion across the faces between a DGGS using an icosahedron as the initial polyhedron and our new DGGS. We show that our

DGGS reduces the mean angular distortion by almost a factor of four while maintaining accurate and efficient queries and without significantly sacrificing desirable DGGS properties. In summary, the main contributions of this work are the creation of a novel DGGS system that preserves area and maintains fast queries while also decreasing angular distortion.

## 2. Related Work

A worldwide collaboration of businesses, government agencies, research organizations and universities called the Open Geospatial Consortium (OGC) was created to facilitate the use and application of geospatial data [8]. The OGC has created a standard for the construction of DGGS and defines desirable properties of DGGS. A considerable body of research work exists to categorize, compare and evaluate DGGS [2,7,17,18]. DGGS are categorized by choice of four components: initial polyhedron, projection, indexing and refinement. Evaluation of DGGS generally takes the form of measuring the distortion of created maps, measuring area and compactness of the created cells, measuring sampling properties of the grid to ensure accuracy and efficiency of statistical calculations, and measuring performance and efficiency of queries of the resulting DGGS. Below, we review research done on each of these components and also look at some recent DGGS.

### 2.1. Initial Polyhedron

As a method for storage and retrieval of data, ideas from existing spatial partitioning data structures (e.g., Quadtree, Octree, R-tree, kd tree and BSP) are often leveraged, but tailored specifically to the spherical space to handle data across the surface of the Earth [17]. All of these structures rely on a hierarchical partitioning of space in order to speed up retrieval operations. This raises the question of how best to partition the Earth for efficient data retrieval. A DGGS extends on the ideas of these spatial partitioning structures where the hierarchy of the resulting cells is used for the multi-resolution representation of data. Early attempts at constructing DGGS used latitude and longitude lines [19]. The cells were constructed by taking discrete steps in latitude and longitude, and multi-resolution could be easily achieved by simple midpoint refinement in each dimension. The advantages of these grids are their simplicity with the straightforward neighbour and hierarchy traversal as well as efficient encoding of latitude/longitude coordinates to the corresponding cells. However, these grids are problematic because the cells do not satisfy either the equal area or the compactness criteria [7]. Additionally, the polar regions of the grid are degenerate with vertices of high valence, reducing compactness in these regions and causing difficulties with data analysis.

In order to address the area and compactness constraints, degenerate (often called igloo) grids have been proposed that still take discrete steps in latitude and longitude, but occasionally change their step size across the grid to produce equal-area cells [20,21]. This creates degenerate areas of the grid at the transition between step sizes, which can complicate neighbour and hierarchy traversal as well as causing similar challenges as with traditional latitude-longitude grids.

Another method for partitioning the Earth uses small circle arcs to construct equal-area cells directly on the sphere [22]. This method avoids the degeneracies of the igloo grids; however, the computational complexity of retrieval and analysis for this type of grid is significantly increased. For example, testing containment of a point inside a cell requires a hierarchical traversal of the grid where at each level, containment within child cells must be tested.

Many recent techniques use a polyhedron whose vertices lie on a sphere to partition the Earth; great circle arcs connect the vertices creating the spherical partition [23,24]. Most of these techniques use platonic solids because they fulfill the area and compactness criteria. Allowing for polyhedra with significantly more faces than the platonic solids is a challenge since some desirable properties of the platonic solids must be sacrificed to allow for more faces. Work has been done to construct grids using geodesic polyhedra with many triangular faces, but it is notoriously difficult to maintain equal area and compactness [25]. How to best distribute vertices across the sphere and define their connectivity in order to satisfy these constraints is an area of ongoing research. A recent work by

Massey uses optimization of a given input partitioning and moves the vertices around on the sphere in an attempt to create compact and equal area cells [25]. In general, these methods can only get close to equal area and produce cells of different shapes and sizes, significantly complicating data retrieval and analysis. Another method by Lee and Mortari uses spherical subdivision starting with an icosahedron in an attempt to construct equal-area compact cells [15]. However, at higher levels of subdivision, this grid produces degeneracies similar to those of igloo grids.

## 2.2. Projection

In order to obtain the benefits of fast analysis of data, we need for there to be a mapping between the spherical grid and a corresponding planar space to allow for more simple analysis using Euclidean methods. Traditionally, this is done using a map projection. Early projections were created to map the sphere to a single continuous plane which produces inevitable distortion [5]. Many current grids opt for a simple projection that does not preserve area but allows for faster data retrieval across the Earth [23–25]. Ideally, these projections *should* preserve the equal area and compactness properties of our original spatial partitioning. In a recent method by Gorski et al., the resulting cells are equal area and the projection used is a combination of the Lambert cylindrical equal-area projection and an interrupted Collignon projection [26]. While preserving area and allowing for multi-resolution across the grid, these two projections introduce significant angular distortion in the planar maps.

As early as 1525, it was proposed that the Earth should be projected piecewise to many planar surfaces corresponding to the faces of a polyhedron, allowing for significantly less distortion [27]. In an unpublished work referenced in 1946, Fisher proposed an equal-area projection to the faces of an icosahedron [28]. While preserving area, this projection produced discontinuities in the map and significant angular distortion. In 1992, Snyder improved on Fisher's work by removing discontinuities and reducing the angular distortion [13]. Snyder's projection also allows for projection onto the faces of a truncated icosahedron (an Archimedian solid), which has more faces than any platonic solid and allows for even less angular distortion. This projection is used in some of the current DGGS because of the desirable sampling properties of the cells and the relatively good distortion properties compared to alternatives [11,29]. However, this projection is computationally expensive and a significant bottleneck to the efficient analysis of data [9,10]. Additionally, there is still significant angular distortion, as can be seen in Figure 2.

To address these challenges, a significant amount of work has been done to develop simple, area-preserving projections for platonic solids. Rosca and Plonka have developed a closed-form, area-preserving projection for the cube and octahedron, respectively [30,31]. In these projections, portions of areas of spherical caps and planar triangles are equated for projection. Due to its simplicity and equal area property, it has been used for the construction of a full DGGS [32].

In work by van Leeuwen and Strebe, two closed-form equal-area projections are proposed that work for any polyhedron with triangular faces [14]. In these projections, an infinitesimal area element is created by dividing a spherical triangle using great and small circle arcs. This infinitesimal area element is then mapped to an infinitesimal planar element using properties of similar triangles. They apply this projection to all the platonic solids and suggest that other polyhedra could be used.

## 2.3. Refinement and Cell Types

Refinement generally depends heavily on the choice of initial polyhedron because only certain refinements work for certain initial cell shapes. Cell shapes generally include triangles, quadrilaterals and hexagons (some grids use irregularly shaped cells not considered here). Mahdavi-Amiri, Alderson and Samavati provide a good survey on refinements and cell types [2]. Triangles are advantageous for being simple to render; quadrilaterals are good for having simple array-based indexing and hexagons have good compactness and sampling properties [33]. Refinement schemes tend to want as few as possible children cells for each parent cell in order to support data at as many different scales as possible. For triangular and quadrilateral shaped cells, 1:4 and 1:9 refinements exist that produce

equal-area planar cells and have reasonably good properties for sampling [24,34,35]. Hexagon cells commonly use a 1:3 refinement but suffer from children not being a partition of their parent [16].

## *2.4. Indexing*

Indexing is chosen for the efficiency of operations on the DGGS [36]. This includes the efficiency of queries such as point, neighbour and hierarchy traversal. There should be a simple relationship between the location in latitude and longitude of a point and the index given to its corresponding cell. This is important because most data comes in the form of latitude and longitude, so having this simple relationship means efficient storage and retrieval of data [37]. For neighbourhood queries, it is also advantageous for the indices of neighbouring cells to be as similar as possible to perform efficient range queries [38]. Retrieving contiguous blocks of memory from a database is efficient and avoids cache misses when retrieving data for neighbouring cells. For hierarchy traversal, there should also be a simple relationship between the index of a cell and its parent and children.

Indexing methods are generally split into three categories: hierarchical, space-filling curve and coordinate-based [2]. In hierarchical indexing, the children of a cell inherit a prefix or a postfix from the index of their parent. The two advantages of this are straightforward hierarchy traversal and simple determination of the resolution of a given cell. For space-filling curves, a curve is passed over the domain of the cells, and the cells are indexed in the order that the curve intersects them (see Figure 3). Often, the curve can be refined to index more fine resolutions of the grid. The advantage of this method is that the curve can be defined to give neighbouring cells similar indices, which allows for efficient range queries. In coordinate based indexing, a coordinate system is defined on the domain of the cells, and the cells are indexed based on taking discrete steps along the coordinate directions. The advantage here is that many operations are efficient and straightforward, but this method works best for regular grids in which the cells are all the same size and shape.

Hybrid methods have also been developed that use a combination of these approaches. An example is Atlas of Connectivity maps that can be used for semi-regular meshes [16]. In this technique, a space-,filling curve can be used for the initial polyhedron while coordinate-based indexing can be used for regular portions of the grid. Many variations of these indexing methods exist; for a full treatment, refer to Reference [36].

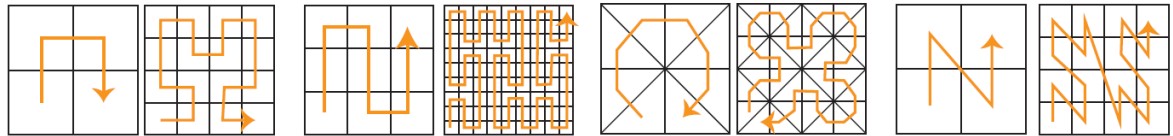

**Figure 3.** From left to right, Hilbert, Peano, Sierpinski and Morton space filling curves. The order that the curve visits the cells determines the indexing. Images taken from [36], licensed under Creative Commons BY 4.0.

## *2.5. DGGS*

For the construction of DGGS, an initial polyhedron, projection, refinement and indexing are generally selected depending on the desired properties of the resulting system. Several works survey and discuss the different choices and their properties [2,8,18].

DGGS have been created from platonic solids as the initial polyhedron since as early as 1968 [39]. Common choices are the cube, octahedron and icosahedron. The cube was historically chosen because its quad faces can be used to naturally extend the quadtree data structure to the surface of Earth [32,40,41]. This allows for efficient storage and retrieval of data. Axis aligning the faces of the cube also allows for simple point queries to determine which face a point resides in. For similar reasons, the octahedron has been chosen because of the simplicity of point queries and because triangles are simple for rendering [34,35,42]. However, these two platonic solids are poor approximations of the sphere and projections defined on them tend to produce significant

distortion [43]. More recent DGGS tend to use the icosahedron as the initial polyhedron for its better distortion properties [43–46]. Many recent DGGS also use equal-area projections to and from the faces of the polyhedron, which necessarily creates angular distortion. A central goal for DGGS has been to define projections that reduce this distortion [43]. Some DGGS also use the truncated icosahedron as their initial polyhedron [11]. The two main advantages are the potential for lower distortion from the increased number of faces and the good sampling properties of hexagonal grids. However, hexagon grids are not ideal for refinement and indexing because children cannot be a partition of their parent, and they are not as simple to render as triangles.

The trend in recent works has been to increase the number of faces in the polyhedron to reduce distortion while deriving an equal-area projection to complete the DGGS. We continue this trend by choosing the DT as our initial polyhedron, which has the most triangular faces of any convex spherical shape with the mirror image property and compact triangles. We then modify the second projection from van Leeuwen and Strebe to work with our initial polyhedron. We complete this work by providing fast and efficient neighbour and hierarchy traversal. Our DGGS maintains the equal area and compactness properties while significantly reducing angular distortion as compared to DGGS using platonic solids and the truncated icosahedron.

## 3. Creating DT DGGS

For reducing the distortion created from projection, we introduce the DT as the initial polyhedron for our DGGS. However, this change requires us to modify other DGGS components in accommodation. We provide these components below.

### 3.1. Initial Polyhedron

In order to reduce angular distortion, we should use an appropriate initial spherical polyhedron—with a higher number of faces than current choices—and a corresponding area-preserving projection. The projection developed by van Leeuwen and Strebe is a good choice because it is efficient and has low angular distortion. However, this projection requires that faces of the initial polyhedron be mirror images of each other along their adjoining edge. Only certain spherical polyhedra can satisfy this criterion. These include the Platonic solids, some of the Catalan solids, and the bipyramids. The bipyramids have the advantage of allowing for any number of faces, but they have poor compactness properties for large numbers of faces (see Figure 4). The Catalan solids allow for more faces than the Platonic solids, so they are the preferred choice.

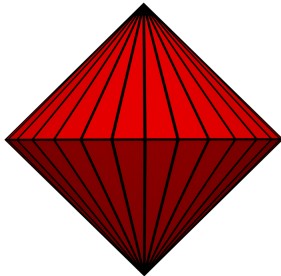

**Figure 4.** A Bipyramid with 64 faces. All faces are identical and mirror images of each other along the adjoining edge but as we increase the number of faces the compactness of the triangles reduces creating longer and skinnier triangles.

Of the Catalan solids, the one with the most faces that best approximates a sphere is the DT—having 120 identical scalene triangle faces—making it the best choice to satisfy our goals (see Figure 1). Since the faces are identical, any projection that is applied individually to each of the faces is guaranteed to be continuous along the edges of the faces because it maps regions identically on

identically shaped faces. Additionally, the faces have equal spherical areas that, when combined with an equal-area projection, satisfy the equal-area criterion. When we use a DT as the initial polyhedron, we lose some compactness of the cells. One measure for the compactness of a spherical region is the Zone Standard Compactness [7], which requires the region's spherical area and perimeter as well as the radius of the sphere. The maximally compact triangle is an equilateral one; the compactness of a DT triangle is about 88% that of an equilateral triangle in an icosahedron (0.727246:0.824773) for a sphere of radius 1.

### 3.2. Equal Area Projection

The properties of the DT make it the ideal candidate for adaptation to the Slice and Dice projection presented by van Leeuwen and Strebe as it has a low angular distortion but requires the mirror symmetry property possessed by the DT [14]. The symmetry and regularity of the mesh ensure that we can have efficient queries, and thus do not lose the speed and simplicity of operations of traditional DGGS.

### 3.3. Slice and Dice Area Preserving Projection

To project points on the Earth to and from points on the DT, the Slice and Dice projection presented by van Leeuwen and Strebe requires some minor modification [14]. As described in Reference [14], in order to map a spherical triangle $ABC$ to a planar triangle $A'B'C'$ (see Figure 5) with constant areal scale, the two triangles are divided into infinitesimal cells. As long as the cells are mapped to each other with a constant areal scale everywhere on the triangles, the mapping is equal area. In the first step of the projection (see Figure 5), a point $P$ within the spherical triangle $ABC$ is to be projected onto an unknown point $P'$ within the planar triangle $A'B'C'$. In order for the projection to be equal area, the spherical triangle is partitioned through the point $P$ and a corresponding partitioning through the projected point $P'$ is created such that areal scale is maintained for the pieces to be projected. To accomplish this, the centre of projection is chosen to be vertex $B$ in the spherical triangle. Great circle arcs emanating from vertex $B$ and crossing great circle arc $AC$ at point $D$ in the spherical triangle will be projected to straight lines emanating from vertex $B'$ and crossing edge $A'C'$ at point $D'$ in the planar triangle. The great circle arc emanating from $B$ through point $P$ cuts the spherical triangle into two parts with areas $T_\text{top}$ and $T_\text{bottom}$. As shown in Reference [14], the spherical and planar divisions are proportioned to the linear parameterization of $A'C'$ where $u'$ and $v'$ are shown in Figure 5:

$$\frac{T_\text{bottom}}{T_\text{top} + T_\text{bottom}} = \frac{T'_\text{bottom}}{T'_\text{top} + T'_\text{bottom}} = \frac{u'}{u' + v'}. \tag{1}$$

In the second step of the equal area projection (see Figure 6), angular distances $x$ and $y$ defined by the point $P$ are solved for from the great circle arc emanating from vertex $B$. Distances $x'$ and $y'$ along our previously constructed line $B'D'$ are determined such that there is constant area scaling for all projected points:

$$\frac{x'^2}{(x' + y')^2} = \frac{1 - \cos(x)}{1 - \cos(x + y)}. \tag{2}$$

Since this is true for all possible points $P$ on the spherical triangle, this means infinitesimal regions bounded by those points will also have the same area.

For triangles in the DT, our procedure has two minor modifications to the method presented by van Leeuwen and Strebe. First, their procedure cuts equilateral triangles in platonic solids into six pieces to be projected separately, but this step is unnecessary for the DT. Because four DT faces correspond to a face of the rhombic triacontahedron (RT), we can use the vertex in the center of each rhombus as the center of projection for each face. Second, they assume a right angle inside of their triangles, allowing for some simplification in the calculation of spherical areas and spherical angles; however, triangles in the DT do not have this right angle.

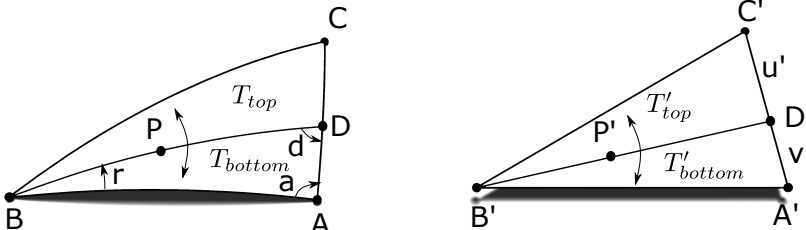

**Figure 5.** Point *P* cuts spherical triangle *ABC* into two parts with a great circle arc emanating from vertex *B*. Point *P′* within a given planar triangle *A′B′C′* is solved for such that the relationship in equation 1 holds.

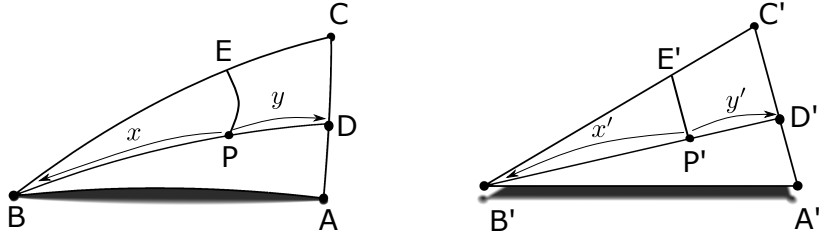

**Figure 6.** Notice that regions *BDC* and *BPE* are both portions of a spherical cap. The area of a spherical cap is proportionate to the angle that it subtends so the area of *BDC* is proportionate to $1 - \cos(x + y)$ and the area of *BPE* is proportionate to $1 - \cos(x)$. Also notice that regions *B′D′C′* and *B′P′E′* are similar triangles. The area of similar planar triangles is proportionate to the square of their side length so the area of the triangle *B′D′C′* is proportionate to $(x' + y')^2$ and the area of *B′P′E′* is proportionate to $x'^2$.

### 3.4. Inverse Projection for the DT

In order to perform all DGGS operations, the projection should include an inverse mapping from points on the polyhedron back to the corresponding point on the sphere. Ideally, this inverse mapping should be efficient and accurate to allow for operations where many points need this performed. Rendering is one such operation where the planar geometry of the cells needs to be inverse projected to transfer data from the polyhedral domain to the corresponding locations on the Earth. At high resolutions, this operation could be required on hundreds of thousands of points. In the publication by van Leeuwen and Strebe, they do not provide the inverse to their projection, and it turns out that it is non-trivial. We have found a closed-form inverse for the Slice and Dice projection, which we present below. To simplify notation, we use $\widehat{XY}$ for the angular distance between $X$ and $Y$, two vertices on a spherical triangle.

For the inverse projection, a given point $P'$ on the planar triangle is mapped to the corresponding point $P$ on the spherical triangle (see Figure 5). Our construction for the inverse mapping is based on reversing steps from the forward projection. We do this by first solving for $D$ such that the ratio in Equation (1) holds, and then solving for $P$ such that the ratio in Equation (2) holds. Our method for finding $D$, and then $P$ is based on the approach from Lee and Mortari [15], where we look for the angular distance $\widehat{AD}$, and $x$ such that we can perform two spherical linear interpolations (SLERP) [47]. First, a SLERP from point $A$ towards point $C$ by an angular distance of $\widehat{AD}$ and then from point $B$ towards point $D$ by an angular distance of $x$, after which we have found point $P$ exactly. In order to accomplish this, we simply need to calculate angular distances $\widehat{AD}$ and $x$.

To start, we solve for the line emanating from $B'$ through $P'$ meeting the planar triangle at point $D'$. Solving for point $D'$ is simply the intersection of two coplanar lines. Now define the ratio of the area of triangles $B'A'D'$ to $A'B'C'$ as

$$m = \frac{\text{area}(B'A'D')}{\text{area}(A'B'C')}. \tag{3}$$

From here, we solve for the great circle arc that cuts the spherical triangle into the corresponding proportionate areas. The solution to a similar problem is presented by Lee and Mortari [15]. However, their solution is limited to the case where $m = 1/2$. Here we extend their approach to solve for point $D$ given $0 < m < 1$. Since $D$ is coplanar to $A$ and $C$ it can be expressed using SLERP with unknown angular distance $\widehat{AD}$ as follows:

$$D = \frac{1}{\sin(\widehat{AC})}(C\sin(\widehat{AD}) + A\sin(\widehat{AC} - \widehat{AD})). \tag{4}$$

In order to find $\widehat{AD}$, we need to find angles $r$ and $d$ from Figure 5. First, notice that by using Girard's spherical excess formula and our triangle area ratio $m$, we can solve for the sum of angles $r$ and $d$ as follows

$$\text{area}(ABC) = a + b + c - \pi. \tag{5}$$

$$\text{area}(ABD) = a + r + d - \pi = m(\text{area}(ABC)). \tag{6}$$

Rearranging we get

$$Y = r + d = m(a + b + c - \pi) - a + \pi. \tag{7}$$

We can express $\widehat{AD}$ using the spherical law of sines in terms of $Y$ and $r$ as follows:

$$\sin(\widehat{AD}) = \frac{\sin(r)\sin(\widehat{AB})}{\sin(Y - r)}. \tag{8}$$

To find $r$, we use the same procedure as Lee and Mortari in Reference [15]. Their derivation is repeated below with our notation. Using the law of spherical cosines on triangle $ABD$ we get

$$\cos(Y - r) = \sin(r)\sin(a)\cos(\widehat{AB}) - \cos(r)\cos(a). \tag{9}$$

Expanding the left-hand side using the angle difference identity for cosine and solving for $r$ we get

$$\tan(r) = \frac{\cos(Y) + \cos(a)}{\sin(a)\cos(\widehat{AB}) - \sin(Y)}. \tag{10}$$

We can now back substitute $Y$ and $r$ to find angular distance $\widehat{AD}$ using Equation (8) and find point $D$ using Equation (4).

Once we have the point $D$ using Equation (4), we now need to find angular distance $x$ to complete the inverse. To do this, we can solve for $\cos(x + y)$ with

$$\cos(x + y) = B \cdot D. \tag{11}$$

Then we can use the known lengths $x'$ and $y'$ as well as $\cos(x + y)$ to solve for $x$ by rearranging Equation (2) to get

$$x = \arccos(1 - (1 - \cos(x + y))\frac{x'^2}{(x' + y')^2})). \tag{12}$$

The position of point $P$ along the great circle $BD$ can be calculated by reusing Equation (4) as follows:

$$P = \frac{1}{\sin(\widehat{BD})}(C\sin(x) + A\sin(\widehat{BD} - x)). \tag{13}$$

In conclusion, we have derived a closed-form inverse projection for the equal-area projection presented by van Leeuwen and Strebe [14]. Our solution relies on simple trigonometric functions without the need for iterative non-linear equation solves, as is the case with the Snyder Equal Area Projection [13]. This allows for efficient rendering of the Earth.

*3.5. Refinement and Indexing*

To define a multi-resolution grid system using the DT, we need a refinement method. There are many different refinement choices for a DGGS, which are detailed in Reference [2]. Figure 7 shows some common choices for refining triangles. Since the initial triangles of a DT are not equilateral, and we wish to preserve area and compactness, it is necessary to use a refinement such as (a), (c) or (d) in Figure 7. The 1:3 refinement shown in (b) is disadvantageous because recursive applications cause an unbounded reduction in compactness as subdivision level increases. Figure 7 (c) and (d) do not have this problem making them more common for DGGS refinement. For both refinements (c) and (d), it is necessary to determine the midpoint of the edges denoted by *M*, *N* and *P*. Notice that the edge denoted by *PN* in refinement (c) is replaced by *MV* in refinement (d), where *M* is the midpoint of the longest edge of the original triangle.

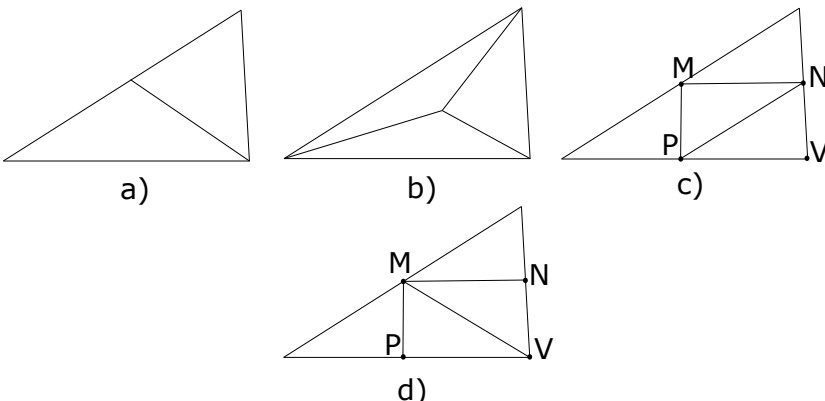

**Figure 7.** Common refinement of triangles includes (**a**) 1:2 (**b**) 1:3 and (**c**) 1:4. Shown in (**d**), a 1:4 composite refinement scheme can also be created by two applications of 1:2 refinement in (**a**).

In order to index our grid system for efficient queries, we use Atlas of Connectivity Maps (ACM) [16]. ACM is a method developed for semi-regular meshes resulting from iterative refinements (e.g., subdivision surfaces). In this method, the irregularity of the base mesh is handled differently from the regular structure resulting from iterative refinement. For example, a half-edge data structure [48] can be used for the irregular connectivity of the base mesh while the regular connectivity resulting from refinement can be handled using simple algebraic relationships detailed in Reference [16].

For the DT grid system, we use a specific scheme for the 30 faces corresponding to faces of the RT (see Figure 8); these 30 faces are indexed in a spiral arrangement. The connectivity between these faces is stored in a $30 \times 30$ array where each entry $(i, j)$ stores the transformation between face $i$ and face $j$ if they are adjacent. For indexing the cells inside each rhombic face, we take advantage of the regular pattern generated by refinement. For example, applying traditional 1:4 refinement to the DT allows for the connectivity map in Reference [16] to be used to efficiently index the resulting cells (see Figure 9). For a composite 1:4 refinement, we propose a modification of row-major ordering. In this method, triangles within rhombuses are indexed using blocks of Z shapes, as illustrated in Figure 10. To differentiate between different subdivision levels, the resolution level of that cell is concatenated to the cell index. This scheme was chosen for its simplicity and the efficiency of the resulting queries.

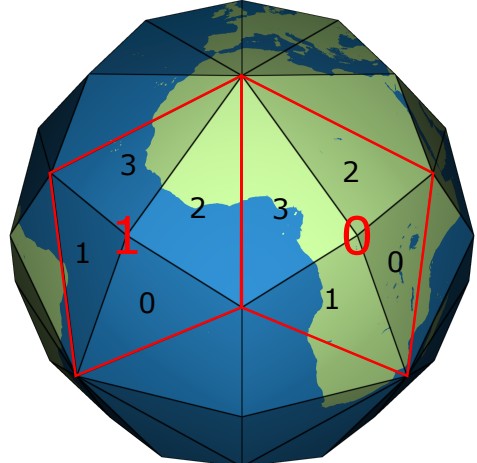

**Figure 8.** The rhombic triacontahedron (RT) is a subset of the vertices and edges of the DT. Two of the rhombic faces and the rhombus indexing are shown in red. The black indexing corresponds to rhombuses in Figure 10.

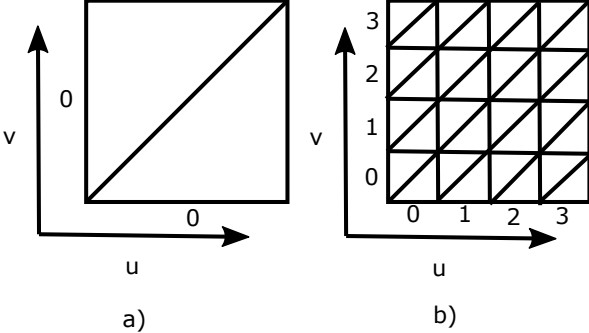

**Figure 9.** (a) Connectivity maps can be assigned to two adjacent triangles to form a quadrilateral domain parameterized by u and v ranging from 0 to 1. (b) Applying traditional 1:4 refinement twice on a triangular connectivity map produces small quad domains within the connectivity map that can be indexed by parameters $u$ and $v$.

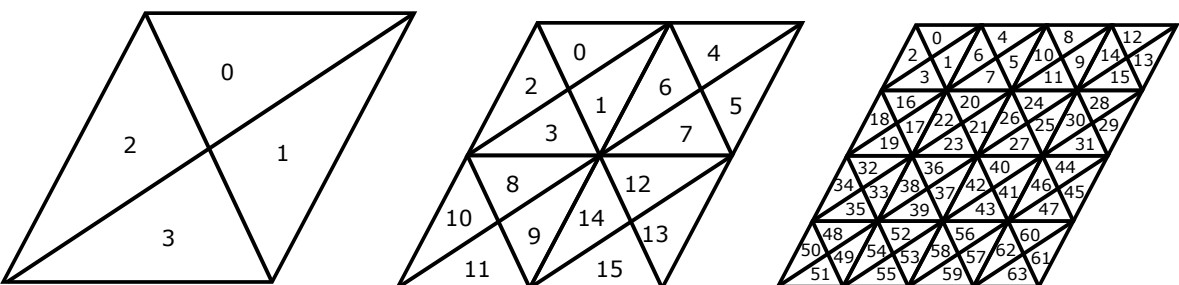

**Figure 10.** An example of the indexing scheme at resolutions 1, 2 and 3 from left to right. Note that this is duplicated for every rhombus in RT. These Z shaped patterns within small rhombuses are repeated along rows and columns of small rhombuses in row major order. We index small rhombuses row by row, most evident in resolution level 3 on the right.

## 4. Connectivity and Location

Since the initial polyhedron of our DGGS has been changed, we need to address important queries such as point-to-cell, neighbourhood access and hierarchy traversal. We show that all of these queries are performed using simple arithmetic in constant time, making them efficient regardless of the resolution of the grid.

### 4.1. Neighborhood Queries

To find the neighbours of a given cell in our grid, we take advantage of the ACM data structure presented in the previous section. Since our initial polyhedron is not degenerate, all triangles have exactly three edge neighbours, and because of our choice of indexing scheme, two of the neighbours are trivial to find. These two neighbours must reside in the small rhombus where the original cell resides (see Figure 11). For finding the third neighbour, the situation is more complicated. First, we must decide if the cell in question is on the boundary of the atlas, which has four cases depending on if the cell in question is on the top, bottom, left or right boundary. If the query cell is on the boundary, we then decide on the relative orientation of the neighbouring cell by checking the ACM data structure that stores the orientation of the neighbours of each base cell (see Figure 8). Depending on the orientation of the neighbouring cells relative to each other, there are two cases (note that for a general mesh, there could be 9 cases, but by construction, we have enforced that our mesh has only two). A linear transformation allows us to change from the coordinate system of the current cell to that of the neighbour. The transformed row and column give us a row and column number in the neighbour base cell, and this gives us the neighbour index. If the cell in question is not on a boundary of the atlas, the situation is much simpler and has four cases corresponding to top, bottom, left and right triangles (see Figure 11). For example, in Figure 11, index five in the small red rhombus has neighbour two in the adjacent base rhombus. These neighbours depend on the relative orientation of the two base rhombuses, which is stored in the ACM data structure.

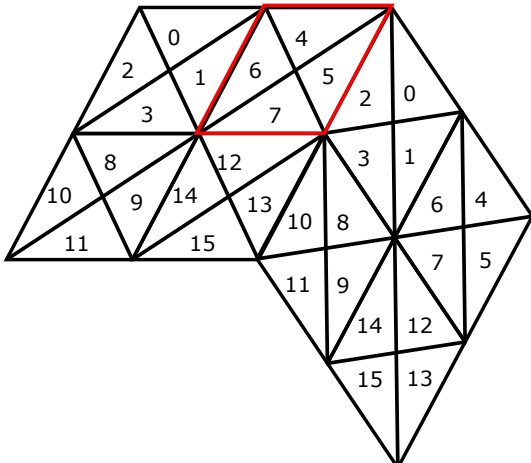

**Figure 11.** Cell indices of two adjacent base rhombuses. Notice that for any cell $c$ in the small rhombus indicated in red, two neighbours are at indices $c \oplus 1$ and $c \oplus 2$. This is true of any index in any small rhombus. The final neighbour depends on if we are on a boundary between rhombuses or not.

### 4.2. Hierarchical Queries

To find the parent of a given cell in the grid, we first find the row and column number of the cell within the base rhombus. For an example of rows and columns, see Figure 12. Next, we find the starting top and bottom index of the small rhombus at one resolution level coarser than the current level (see the green labels in Figure 12). After that, we solve for the parent index given the row and column solved for previously (note that parents have half the number of rows and columns). Based on the index inside the small rhombus at one resolution level coarser than the current level, we can solve for which parent cell using 16 cases shown if Figure 13. Children of a cell are calculated using the reverse of this process.

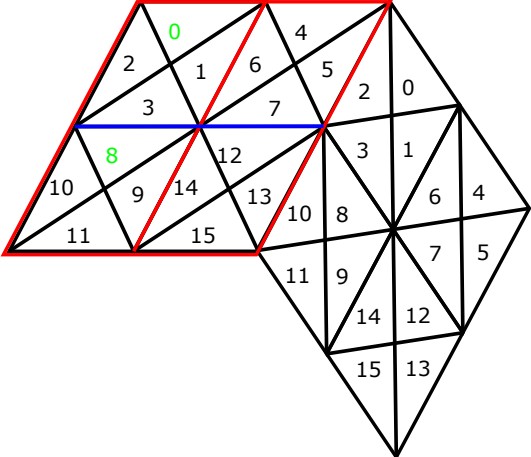

**Figure 12.** The rhombus highlighted in red will become a parent rhombus when we decrease subdivision levels. Rows are divided with a blue line—columns with a red line. The green indices are the start of two consecutive blocks of indices within the red rhombus. Specific indices become part of specific parents as seen in Figure 13.

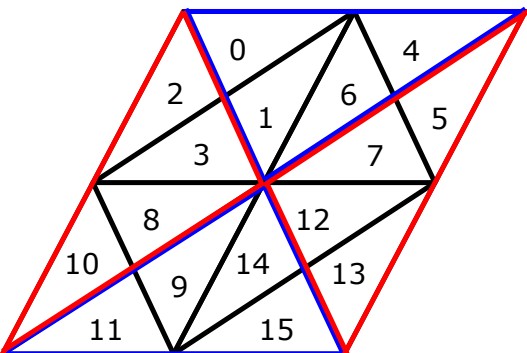

**Figure 13.** Parent cells are highlighted in blue and red. Note that 16 child cells make up the four parents in the rhombus.

### 4.3. Point-to-Cell Query

To find the index of a cell given a point in latitude-longitude coordinates, we first find the triangle in the DT that contains the point. To speed up this part of the query, we have aligned the DT with the longitude axis such that a vertex of valence ten lies on both the North and South pole (see Figure 14). We align the edges emanating from these two vertices to be aligned with multiples of $2\pi/10$ radians of longitude.

For finding the cell containing the given point $P$, we first determine which face $F$ of the DT that contains $P$ (see Figure 14); then, we find the child of $F$ at a specific resolution that contains $P$. To find $F$, merely checking the longitude of the point allows us to narrow our search to 12 possible triangles. To distinguish between the remaining 12, we check if the point lies above or below the edges of these triangles. This is done in using a binary search in which a maximum of 5 edges need to be tested to determine the triangle uniquely.

Next, for finding the children of $F$ at the resolution $s$ containing $P$, we project our point from the sphere to $F$ using our equal-area projection scheme detailed in Section 3.2. After that, we express our projected point $P'$ as barycentric coordinates ($u$ and $v$) of the planar triangle $F$, which will give us the row and column number of the cell index that contains the point. Once we have the row and column, we do a final check to determine which of the four triangles in that row and column contains the point $P$. To do this, we first find the number of rows and columns $n$ from the resolution $s$ with

$$n = 2^{s-1}. \tag{14}$$

We then find the row $r$ and column $c$ of the point:

$$r = \lfloor v * n \rfloor, \quad \text{and} \tag{15}$$

$$c = \lfloor u * n \rfloor. \tag{16}$$

Since the planar triangle is evenly subdivided into rows and columns, this gives us the small rhombus containing the point. To figure out which triangle contains the point, we find a row and column offset, which is the truncated fractional component from the floor operation above. This gives us an offset into the small rhombus. We then test this offset against the functions $y = x$ and $y = 1 - x$ to determine the triangle. We define $o$ as the calculated triangle number of the four within the small rhombus. Our final triangle index $i$ is then

$$i = 2^{s+1} * r + (4 * c) + o. \tag{17}$$

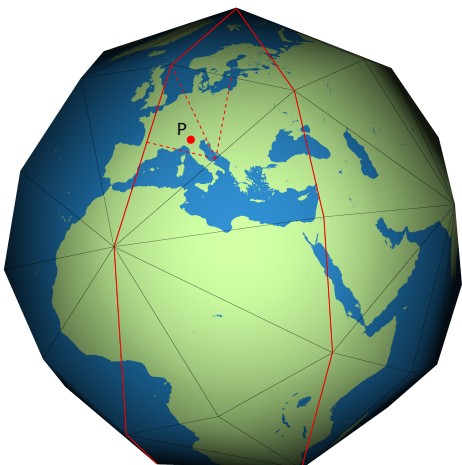

**Figure 14.** A DT with great circle arcs (longitude) between the North and South pole highlighted in red. The region between these arcs contains 12 base triangles. A point $P$ on the Earth can be quickly checked to see if it lies within the partition created by the red arcs by simply checking its longitude. The dashed lines represent the next subdivision level. The barycentric coordinates of the point within the base triangle determines which child cell $P$ lies in.

## 5. Results

For evaluation of our new DGGS, we compare the angular distortion of Slice and Dice projection onto faces of a DT to Slice and Dice projection onto faces of an icosahedron DGGS. To measure angular distortion, we use the fact that an infinitesimal circle on the Earth will be projected to an infinitesimal ellipse on our planar cell. The major and minor axis ($a$ and $b$) of the projected ellipse quantifies the amount of angular distortion $d$ created by the projection according to the following formula [14].

$$d = 2 * \arcsin \frac{a - b}{a + b} \tag{18}$$

Geometrically, the quantity $d$ is the maximum difference in angle (from $-\pi$ to $\pi$) between a point on a circle and the corresponding point on the ellipse with major axis $a$ and minor axis $b$. The correspondence between the circle and the ellipse is made by the scaling transformation that maps the circle onto the ellipse. This change in angle is the local change in bearing necessary to navigate between points on the planar map.

To estimate the distortion created by Slice and Dice projections, we distribute many tiny circles across our spherical triangle (see Figure 15). We project these circles onto the planar triangle and find

our major and minor axes. Using $d$ in equation 18 provides an estimate of the distortion at all the sampled circle centers. The mean and standard deviation of the angular distortion for both the DT and the icosahedron can be found in Table 1.

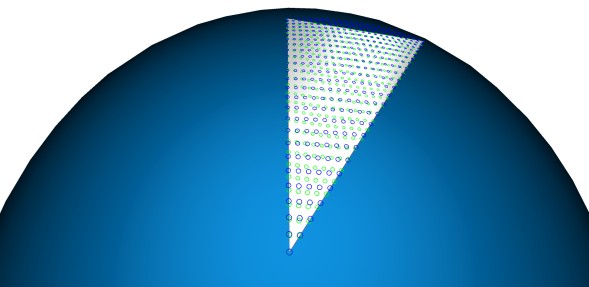

**Figure 15.** Circles on the spherical face of a DT shown in blue are projected to ellipses on the planar face of a DT shown in green. We can see that the angular distortion is minor because the major and minor axis of the ellipses are close to the same.

**Table 1.** Mean and standard deviation of angular distortion (in radians) of the projection to the icosahedron and the DT. The mean for the DT is approximately four times less and the standard deviation is lower showing that projection to the DT has significantly reduced angular distortion.

|  | Mean Angular Distortion (Radians) | Standard Deviation |
|---|---|---|
| Icosahedron | 0.144 | 0.027 |
| Disdyakis Triacontahedron | 0.039 | 0.016 |

To visualize the distribution of the distortion, we have mapped the absolute values of $d$ to greyscale values shown in Figure 16. In this figure, we have mapped the maximum absolute value of $d$ found in the icosahedron to white and the minimum absolute value of $d$ found in the icosahedron to black. We have used the same range of distortion values to greyscale mapping for both the icosahedron and the DT; we see in Figure 16 that the entire face of the DT is black indicating a significant improvement.

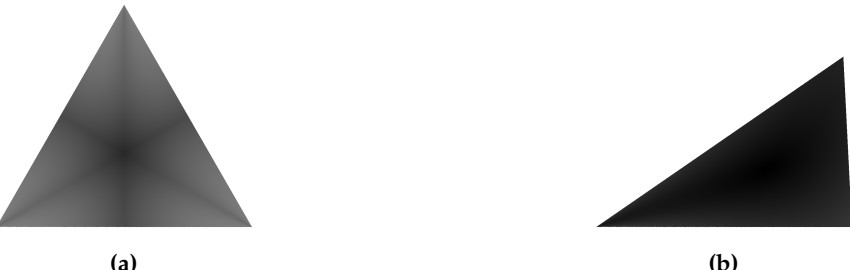

**(a)**                                        **(b)**

**Figure 16.** (**a**) Angular distortion mapped to a greyscale value across a face of an icosahedron. (**b**) Angular distortion mapped to a greyscale value across a face of a DT. Black is low distortion and white is high distortion. We have used the same greyscale mapping for both for direct comparison.

Figure 17 demonstrates the output of our system and shows the impact of low angular distortion on the resulting grid geometry. The resolution of this grid is five, and it partitions the Earth into 30,720 triangles.

Additionally, we have implemented a fully functional DGGS using the proposed methods. Figure 18 shows a visualization of temperature across the Earth using two years of monthly data, linearly interpolated between months and bilinearly interpolated onto the vertices of the DT. We can query and visualize this data across the entire Earth in real-time.

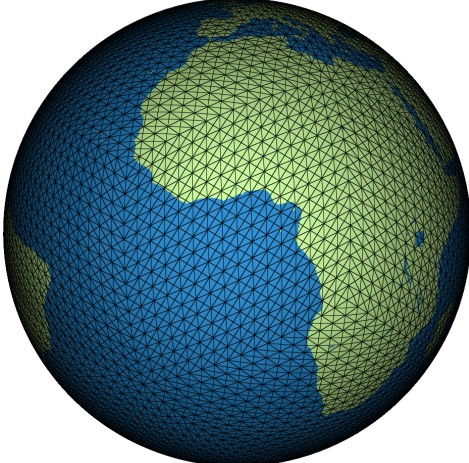

**Figure 17.** DT subdivided to resolution 5. The angular distortion along edges is not visible.

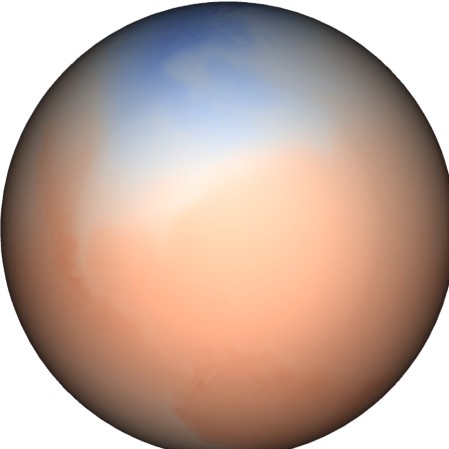

**Figure 18.** Temperature data interpolated across the DT Discrete Global Grid System (DGGS). Cold areas are blue and red areas are hot. This data can be queried and visualized across the Earth in real time.

## 6. Conclusions

In this paper, we have introduced a new equal-area DGGS using the DT. To support this new initial polyhedron, we have extended the Slice and Dice projection to have a closed-form inverse and proposed refinement and indexing methods for this grid system. Our new DGGS reduces angular distortion as compared to other DGGS while maintaining area preservation at all levels of resolution. We have shown that our new DGGS can be implemented to support fast and efficient queries across the surface of the Earth.

*Future Work*

There are a number of future directions to explore. First, further research into other polyhedra may provide viable alternatives here. Allowing the user to specify the tradeoff between different properties for a particular problem or application would allow the best possible polyhedron for each use case. We would also like to explore different indexing schemes to help with memory locality for neighboring cells. Our indexing scheme traded efficiency of neighborhood queries for memory locality. Especially as resolutions become high, cache misses during the queries could become a substantial bottleneck to the performance of the system, and different choices for the indexing scheme may allow for a method that has better storage properties. Another remaining question is whether or not we can find grids with equal area cells that have more faces than the DT for further reducing distortion.

It is an open research question of how to construct equal-area grids on the sphere without producing degenerate regions.

**Author Contributions:** Conceptualization, John Hall and Benjamin Ulmer and Faramarz Samavati; Data curation, Lakin Wecker; Formal analysis, John Hall and Lakin Wecker and Benjamin Ulmer; Funding acquisition, Faramarz Samavati; Investigation, John Hall and Lakin Wecker and Benjamin Ulmer; Methodology, John Hall and Lakin Wecker and Benjamin Ulmer; Project administration, Faramarz Samavati; Software, John Hall and Lakin Wecker and Benjamin Ulmer; Supervision, Faramarz Samavati; Validation, John Hall and Lakin Wecker and Benjamin Ulmer; Visualization, John Hall and Lakin Wecker and Benjamin Ulmer; Writing—original draft, John Hall; Writing—review and editing, John Hall and Lakin Wecker and Benjamin Ulmer and Faramarz Samavati. All authors have read and agreed to the published version of the manuscript.

**Acknowledgments:** Special thanks to all members of the GIV research team at the University of Calgary for insights and discussion.

**Funding:** This research was partially funded by the Natural Sciences and Engineering Research Council of Canada (NSERC).

**Conflicts of Interest:** The authors declare no conflict of interest.

## Abbreviations

The following abbreviations are used in this manuscript:

DE      Digital Earth
DGGS    Discrete Global Grid System
DT      Disdyakis Triacontahedron
RT      Rhombic Triacontahedron

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
