# Peer review of "Disdyakis Triacontahedron DGGS"

_ijgi, doi:10.3390/ijgi9050315_

Round 1
Reviewer 1 Report
This is a nice work on hierarchical spherical grids, using an equal area projection from the Disdyakis Triacontahedron. Since DT has 120 faces, the angular distortions are smaller than in the previous constructions, which use polyhedrons with less faces.
The construction presented here is based on the "Slice and Dice" projection from the sphere to a polyhedron, realized by Leeuwen and Strebe in [14]. The authors also construct the inverse map, from the DT to the sphere, allowing the whole construction of hierarchical spherical grids. An efficient indexing procedure for the hierarchical cells is also presented.
Since I do not have Mathematica at home, where I am forced to stay, I would like to see a mathematical proof (a Mathematica program or by hand) of the following:
- the inverse projection deduced here, composed by the initial map indeed maps a point P into itself - I would like to see calculations using mathematical formulas, not a geometric argument.
- a Jacobian equal to 1, showing that the mapping (or its inverse) is indeed an equal area projection map
This can be included in the manuscript, if it is not too long, but for me is a must, to check the accuracy of the formulas.
Abstract: replace anglar -> angular.
Reviewer 2 Report
The article is relevant, contains new scientific results, convincingly reasoned and clearly shown, and is relevant - the DGGS format and Digital Earth are both actively investigated. The results have both academic and practical applications.
Author Response
Response to Reviewer 2 Comments
No comments to address.
Reviewer 3 Report
- Paper is well researched and demonstrates good knowledge of the literature.
- Illustrations are helpful and well chosen.
- Clarity and organization are good.
- Closed-form inverse for the projection is a useful contribution.
- Indexing scheme shows good attention to the relevant concerns of a DGGS.
- Results are an important advancement in DGGSs: Without this work, the disdyakis triacontahedron would not be an accessible choice for a DGGS. With this work, the convex solid with the highest uniform face count can now be used for indexing the sphere.
- Please state units in Table 1 (degrees or radians?).
- Leeuwen is properly van Leeuwen. (Please confirm.)
- Comparisons to distortion in other DGGS besides icosahedron would be useful. (Not required for publication.)
Author Response
Response to Reviewer 3 Comments
1.Table 1 has been updated with units (radians in this case).
2.Leeuwen is properly van Leeuwen and has been updated at every occurence in the paper.
3.Comparison to other DGGS would indeed be useful but was not within the scope of this work. However, in the Slice and Dice paper distortion of this method is compared to all other platonic solids and to Snyder Projection also across all the Platonic solids which gives a reasonable idea of the distortion when we change polyhedron and when we change projection.